# Sphingosine-1-Phosphate Receptor 2 Controls Podosome Components Induced by RANKL Affecting Osteoclastogenesis and Bone Resorption

**DOI:** 10.3390/cells8010017

**Published:** 2019-01-01

**Authors:** Li-Chien Hsu, Sakamuri V. Reddy, Özlem Yilmaz, Hong Yu

**Affiliations:** 1Department of Oral Health Sciences, College of Dental Medicine, Medical University of South Carolina, Charleston, SC 29425, USA; vincentlichien@gmail.com (L.-C.H.); yilmaz@musc.edu (Ö.Y.); 2Department of Pediatrics, Osteoclast Center, Darby Children’s Research Institute, Medical University of South Carolina, Charleston, SC 29425, USA; reddysv@musc.edu

**Keywords:** sphingosine-1-phosphate receptor 2, cytokines, osteoclast, chemotaxis, podosome, bone loss

## Abstract

Proinflammatory cytokine production, cell chemotaxis, and osteoclastogenesis can lead to inflammatory bone loss. Previously, we showed that sphingosine-1-phosphate receptor 2 (S1PR2), a G protein coupled receptor, regulates inflammatory cytokine production and osteoclastogenesis. However, the signaling pathways regulated by S1PR2 in modulating inflammatory bone loss have not been elucidated. Herein, we demonstrated that inhibition of S1PR2 by a specific S1PR2 antagonist (JTE013) suppressed phosphoinositide 3-kinase (PI3K), mitogen-activated protein kinases (MAPKs), and nuclear factor kappa-B (NF-κB) induced by an oral bacterial pathogen, *Aggregatibacter actinomycetemcomitans*, and inhibited the release of IL-1β, IL-6, TNF-α, and S1P in murine bone marrow cells. In addition, shRNA knockdown of S1PR2 or treatment by JTE013 suppressed cell chemotaxis induced by bacteria-stimulated cell culture media. Furthermore, JTE013 suppressed osteoclastogenesis and bone resorption induced by RANKL in murine bone marrow cultures. ShRNA knockdown of S1PR2 or inhibition of S1PR2 by JTE013 suppressed podosome components, including PI3K, Src, Pyk2, integrin β_3,_ filamentous actin (F-actin), and paxillin levels induced by RANKL in murine bone marrow cells. We conclude that S1PR2 plays an essential role in modulating proinflammatory cytokine production, cell chemotaxis, osteoclastogenesis, and bone resorption. Inhibition of S1PR2 signaling could be a novel therapeutic strategy for bone loss associated with skeletal diseases.

## 1. Introduction

Local or systemic bone loss occurs in many human diseases, including rheumatoid arthritis, systemic lupus erythematosus, axial spondyloarthritis, psoriatic arthritis, inflammatory bowel disease, postmenopausal osteoporosis, and periodontitis [1]. These diseases are characterized by high levels of proinflammatory cytokines, such as interleukin (IL)-1β, IL-6, tumor necrosis factor (TNF)-α, and receptor activation of nuclear factor kappa-B ligand (RANKL) [1,2,3,4,5]. Additionally, inflammatory conditions are associated with high levels of a bioactive sphingolipid, sphingosine-1-phosphate (S1P) [6,7,8]. IL-1β, IL-6, TNF-α, and S1P promote chemotaxis of monocytes (osteoclast precursors) from blood circulation to bone tissues [8,9,10,11], while RANKL, IL-1β, IL-6, and TNF-α stimulate the differentiation and fusion of monocytes and macrophages to form multinucleated osteoclasts, leading to bone loss [12,13].

S1P is generated from sphingosine by activation of sphingosine kinase (SK) 1 and / or 2 by various stimuli including bacterial lipopolysaccharides (LPS) and cytokines [14,15]. S1P can be degraded by S1P lyase or dephosphorylated by S1P phosphatase [15,16]. Constitutive levels of S1P in most tissues are very low (10–30 nM) [15,16] because S1P is either degraded by S1P lyase or dephosphorylated by S1P phosphatase in tissues. In contrast, S1P levels in the blood are very high (150–1000 nM) [15,16] because erythrocytes and platelets generate abundant S1P, but erythrocytes and platelets lack both S1P lyase and S1P phosphatase [15,16]. As a result, there is a sharp S1P gradient between the blood and tissues, which controls the migration of monocytes from blood to tissues, affecting various immune responses [15,16].

Lee et al. [7] showed that postmenopausal women had higher S1P plasma levels in comparison to premenopausal women and men. In these postmenopausal women, the high S1P plasma levels were positively correlated with low bone mineral density [7]. Our previous study [8] showed that S1P dose-dependently increased chemotaxis of murine bone marrow-derived monocytes and macrophages (BMMs). We also showed that BMMs derived from SK1 deficient mice reduced S1P generation induced by an oral bacterial pathogen, *Aggregatibacter actinomycetemcomitans* (*Aa*), and that SK1 deficiency in mice alleviated periodontal alveolar bone loss induced by *Aa* [8]. It also has been shown that the synovial fluid of patients with rheumatoid arthritis exhibited significantly higher levels of S1P than their non-inflammatory osteoarthritis counterparts [17]. Genetic SK1 deficiency in mice significantly decreased synovial inflammation and joint erosions in murine TNF-α-induced arthritis [18].

S1P receptor 2 (S1PR2), also called endothelial differentiation G-protein coupled receptor 5 (EDG5), is one of the five G protein-coupled S1P receptors (S1PR1–5). S1PR2 is expressed in most tissues and on the plasma membrane of mammalian cells [19,20]. S1PR2 couples with G_i_, G_q_, and G_12/13_ family proteins and modulates Rac, Rho, phospholipase C (PLC), phosphoinositide 3-kinase (PI3K), nuclear factor kappa-B (NF-κB), and mitogen-activated protein kinases (MAPKs) [19,20,21,22,23,24]. MAPKs include extracellular signal-regulated kinase (ERK), c-Jun N-terminal kinase (JNK), and p38 MAPK. 

Ishii et al. [25] demonstrated that S1PR2 inhibited the chemotaxis of BMMs. They showed that treatment with a specific S1PR2 siRNA increased S1P-induced chemotaxis of BMMs. Moreover, wild type mice treated with a specific S1PR2 antagonist (JTE013) changed monocyte migration behavior induced by RANKL by enhancing monocyte percentage in the blood and alleviated osteoporosis induced by RANKL [25]. 

Our previous study [23] demonstrated that S1PR2 played an important role in regulating proinflammatory cytokine release induced by the oral bacterial pathogen *Aa*. Lentiviral delivery of S1PR2 shRNA significantly reduced IL-1β, IL-6, and TNF-α protein levels induced by *Aa* in BMMs compared with controls. Mechanistically, we demonstrated that knockdown of S1PR2 suppressed p-PI3K, p-ERK, p-JNK, p-p38 MAPK, and p-NF-κBp65 protein levels induced by *Aa*. In addition, we demonstrated that S1PR2 played a critical role in regulating osteoclastogenesis induced by RANKL [23]. Knockdown of S1PR2 by the S1PR2 shRNA inhibited osteoclastogenesis and suppressed bone resorption in murine bone marrow cells induced either by RANKL alone or co-stimulation by RANKL and *Aa*-stimulated cell culture media compared with controls [23]. We further showed that knockdown of S1PR2 significantly suppressed factors associated with osteoclast formation/activity, including the nuclear factor of activated T-cells cytoplasmic calcineurin-dependent 1 (Nfatc1), cathepsin K (Ctsk), acid phosphatase 5 (Acp5), osteoclast-associated receptor (Oscar), dendritic cell-specific transmembrane protein (Dc-stamp), and osteoclast stimulatory transmembrane protein (Oc-stamp) induced by RANKL in bone marrow cells compared with controls [23]. However, the signaling pathways regulated by S1PR2 in modulating RANKL-induced osteoclastogenesis have not been elucidated. 

Osteoclasts, the multinucleated bone resorption cells, are formed by fusion of monocytes and macrophages [26]. Osteoclast adhesion involves structural units called podosomes [27]. Podosomes are composed of a filamentous actin (F-actin) core surrounded by a ring structure containing protein kinases (PI3K, Src, and Pyk2), integrins (β_1_, β_2_, β_3_, α_M_β_2_, α_v_β_3_), and integrin-associated proteins (including paxillin) [27]. During osteoclastogenesis, podosomes cluster into a podosome belt, followed by formation of a sealing zone. The sealing zone structure creates an isolated resorption compartment between osteoclasts and bone matrix, which contributes to bone resorption [28,29,30]. 

In this study, we determined if pharmacological inhibition of S1PR2 by its specific antagonist (JTE013) can inhibit IL-1β, IL-6, TNF-α, and S1P production; suppress chemotaxis of BMMs; and attenuate osteoclastogenesis and bone resorption, serving as a potential novel therapeutic strategy for inflammatory bone loss diseases. Additionally, we delineated the role of S1PR2 in regulating podosome components (including PI3K, Src, Pyk2, F-actin, integrin β_3_, and paxillin) induced by RANKL.

## 2. Materials and Methods

### 2.1. Animals, Cells, and Reagents

Six to eight-week-old C57BL/6J mice were purchased from Jackson Laboratory (Bar Harbor, ME, USA). Bone marrow (BM) cells were harvested from mice by flushing BM with complete minimal essential media (MEM)-α (Life Technologies, Grand Island, NY, USA), supplemented with 10% fetal bovine serum (FBS), 100 U/mL penicillin, and 100 µg/mL streptomycin. BM-derived monocytes and macrophages (BMMs) were cultured as previously described [23]. JTE013 was purchased from Cayman Chemical (Ann Arbor, MI, USA), dissolved in ethanol, and diluted in serum-free MEM-α media. Vehicle (ethanol) and JTE013 (2 to 8 µM) were used. The animal study was performed in accordance with ARRIVE guidelines for animal research. All animal procedures used in this study were approved by the Institutional Animal Care and Use Committee (IACUC) at the Medical University of South Carolina (MUSC). 

### 2.2. Generation of shRNA Lentivirus 

Murine S1PR2 shRNA lentiviral vector and a control shRNA lentiviral vector were generated as previously described [23]. The viral pellet was resuspended in serum-free DMEM medium, and viral titer was determined by a HIV-1 p24 antigen ELISA kit (Zeptometrix, Buffalo, NY, USA). In this study, BMMs were infected with lentiviral vector at 25 multiplicity of infection (MOI). 

### 2.3. Culture of Aggregatibacter Actinomycetemcomitans 

*Aggregatibacter actinomycetemcomitans* (*Aa*, ATCC 43718) was purchased from American Type Culture Collection (Manassas, VA, USA). Bacteria were cultured and bacterial concentration was quantified as previously described [23]. In this study, BMMs were infected with 1.5 colony forming unit (CFU)/cell of *Aa*. *Aa*-stimulated cell culture media (*Aa*-media) was obtained by filter-sterilization of cell culture media derived from BMMs infected with *Aa* for 6 h. 

### 2.4. Enzyme-linked Immunosorbent Assay (ELISA)

IL-1β in cell lysates, IL-6, and TNF-α protein levels in cell culture media of BMMs were quantified by ELISA kits (R&D Systems, Minneapolis MN, USA). The concentration of cytokines was normalized by protein concentration, which was determined by a DC protein Assay Kit (Bio-Rad Laboratories, Hercules, CA, USA) in cell lysates. 

### 2.5. Mass Spectrometry Analysis of Sphingolipids 

Sphingolipids were extracted from the cell protein lysates or cell culture media by the Lipidomics Shared Resource at MUSC, using the Bligh Dyer technique. Sphingolipid analysis was performed using a Thermo Finnigan TSQ 7000 triple quadruple mass spectrometer. This technique has been previously described by Bielawski et al. [31].

### 2.6. Cell Viability Assay

BM cells (1 × 10^5^/well) in a 96-well plate were incubated with JTE013 (2 to 8 μM) or control vehicle ethanol for 24 h. Cell viability was analyzed by CellTiter 96 Aqueous One Solution Cell Proliferation Assay (Promega, Madison, WI, USA).

### 2.7. Transwell Chemotaxis Assay

1 × 10^5^ of BMMs, treated with (1) S1PR2 shRNA, (2) control shRNA, (3) JTE013, or (4) vehicle, were put in the upper chambers of transwell plates (6.5 μM, Corning Incorporated, Corning, NY, USA), respectively, in MEM-α media with 1% FBS. The lower chambers were filled with either (1) serum-free MEM-α media, (2) media derived from BMMs treated with S1PR2 shRNA and infected with *Aa* for 6 h, (3) media derived from BMMs treated with control shRNA and infected with *Aa* for 6 h, (4) media derived from BMMs treated with JTE013 and infected with *Aa* for 6 h, and (5) media derived from BMMs treated with vehicle and infected with *Aa* for 6 h, respectively. After 6 h of incubation, the inserts were fixed with 10% glutaraldehyde for 10 min and stained with 2% crystal violet for 20 min at room temperature (RT). After washing inserts in water for 4 s, the cells on the top of inserts were removed by wiping off with cotton swabs. The inserts were dried and mounted on slides with coverslips. The number of cells in 10 fields of 400× magnification views was quantified by light microscopy. The average number of cells per 400× magnification view served as migration index.

### 2.8. Western Blot Analysis 

Protein was extracted from BMMs by RIPA cell lysis buffer (Cell signaling Technology, Danvers, MA, USA). Total protein (30 μg) was loaded on 10% Tris-HCl gels, electro-transferred to nitrocellulose membranes, blocked, and incubated overnight at 4 °C with primary antibody. The antibodies to p-PI3K, p-ERK, p-JNK, p-p38, p-NF-κB p65, p-Src, p-Pyk2, integrin β3, and glyceraldehyde-3-phosphate dehydrogenase (GAPDH) were purchased from Cell Signaling Technology (Danvers, MA, USA). Antibody to F-actin was obtained from Abcam (Cambridge, MA, USA). An antibody to paxillin was purchased from Santa Cruz Biotechnology Inc. (Dallas, TX, USA). All primary antibodies were used at 1:1000 dilution. After washing, the nitrocellulose membranes were incubated at RT for 1 h with horseradish peroxidase-conjugated secondary antibodies (Cell Signaling Technology) and developed using SuperSignal West Pico Chemiluminescent Substrate (Life Technologies Grand Island, NY, USA). Digital images and protein densitometry were analyzed with a G-BOX chemiluminescence imaging system (Syngene, Frederick, MD, USA). 

### 2.9. Osteoclastogenesis Assay and Tartrate-Resistant Acid Phosphatase (TRAP) Staining

Murine BM cells were cultured for three days in complete MEM-α media supplemented with 50 ng/mL recombinant murine M-CSF to allow BM progenitor cells to proliferate and differentiate. The BM cells were plated in new culture dishes. BM cells were either treated with vehicle (ethanol) or JTE013 (8 µM) and were cultured in complete MEM-α media containing both M-CSF (50 ng/mL, R&D systems) and RANKL (250 ng/mL, R&D Systems) to allow BM cells to differentiate into pre-osteoclasts. A control group of cells were cultured with only M-CSF. The cell culture media was changed at 48 h and 72 h. After 72 h, some of the cells were stimulated with *Aa*-stimulated culture media (200 µL/mL) alone or co-stimulated with RANKL and *Aa*-stimulated culture media for 24 h. Four days after RANKL treatment, TRAP staining was performed in cultured BM cells using a leukocyte acid phosphatase kit (Sigma Aldrich, St. Louis, MO, USA). Pictures were taken using a Nikon Eclipse TS-100 inverted microscope. Image analysis was performed using Visiopharm 5.0 software (Visiopharm, Hoersholm, Denmark). 

### 2.10. Bone Resorption Assay

Murine BM cells were cultured for three days in complete MEM-α media supplemented with 50 ng/mL recombinant murine M-CSF and plated in a calcium phosphate-coated 48-well plate (Cosmo Bio USA, Carlsbad, CA, USA). Cells were cultured in complete MEM-α media containing both M-CSF (50 ng/mL) and RANKL (500 ng/mL), with vehicle (ethanol) or JTE013 (8 µM). A control group of cells was cultured with only M-CSF. On the third and fifth day, the cell culture media was changed with or without RANKL and/or JTE013. Some of the cells were stimulated with *Aa*-stimulated culture media (200 µL/mL) alone or co-stimulated with RANKL and *Aa*-stimulated culture media. Seven days after treatment, cells were removed by treatment with 5% sodium hypochlorite for 5 min. After washing and drying of the plate, bone resorption pit images were taken by a Nikon Eclipse TS-100 inverted microscope and analyzed by Visiopharm 5.0 software. 

### 2.11. RNA Extraction and Real Time PCR

Total RNA was isolated from cells using TRIZOL (Life Technologies) according to the manufacturer’s instructions. Complementary DNA was synthesized by a TaqMan reverse transcription kit (Life Technologies) using the total RNA (1 μg). Real-time PCR was performed using a StepOnePlus Real-Time PCR System (Life Technologies). PCR conditions used were as follows: 50 °C for 2 min, 95 °C for 10 min, and 40 cycles of 95 °C for 15 s, 60 °C for 1 min. The following amplicon primers were obtained from Life Technologies: Nfatc1 (Mm00479445_m1), Ctsk (Mm00484039_m1), Acp5 (Mm00475698_m1), Oscar (Mm00558665_m1), Dc-stamp (Mm04209236_m1), Oc-stamp (Mm00512445_m1), RANKL (Mm00441906_m1), OPG (Mm01205928_m1), and GAPDH (Mm99999915_g1). The mouse S1PR2 primers (PrimePCR^TM^ SYBR^®^ Green Assay) were obtained from Bio-Rad Laboratories (Hercules, CA, USA). Amplicon concentration was determined using threshold cycle values compared with standard curves for each primer. Sample mRNA levels were normalized to an endogenous control GAPDH expression and were expressed as fold changes as compared with control groups. 

### 2.12. Immunofluorescence 

BMMs on coverslip were fixed by 4% paraformaldehyde for 10 min at RT. After three washings with PBS, the cells were permeabilized by treatment with 0.1% Triton X-100 in PBS for 10 min at RT. After another three washings with PBS, cells were blocked with 5% goat serum for 1 h at RT. Following another three washings with PBS, the cells were incubated with anti-integrin β3 (1:100, Cell Signaling Technology) or anti-paxillin (1:60, Santa Cruz Biotechnology Inc.) with 1% BSA overnight at 4 °C. Cells were washed again with PBS, and incubated with an Alexa fluor 488 labeled secondary antibody (1:400, Fisher Scientific, Suwanee, GA, USA) with 1% BSA for 1 h at RT. After washing with PBS, cells were incubated with rhodamine phalloidin (1:140, Cytoskeleton, Inc., Denver, CO, USA) for 1 h at RT. Cells were washed again with PBS and mounted on a slide with Prolong Gold anti-fade reagent with DAPI (Fisher Scientific) overnight at RT. Digital images were recorded by an Olympus BX61fluorescent microscope. Fluorescence intensity analysis was performed using 20 cells per image. Fluorescence intensity was quantified by Adobe Photoshop using the following formula: corrected cell fluorescence = integrated density − (area of selected cell × mean fluorescence of background).

### 2.13. Statistical Analysis

All experiments were performed in triplicate with BM cells from mice. Data were analyzed by one-way ANOVA with Tukey multiple comparisons test. All statistical tests were performed using GraphPad Prism software (GraphPad Software Inc., La Jolla CA, USA). Values are expressed as means ± standard error of the means (SEM) of three independent experiments. A *p* value of 0.05 or less was considered significant. 

## 3. Results

### 3.1. Inhibition of S1PR2 by Its Specific Antagonist (JTE013) Reduced IL-1β, IL-6, TNF-α, and S1P Levels Induced by Aa in BMMs

Our previous study [23] demonstrated that knockdown of S1PR2 by a specific S1PR2 shRNA lentiviral vector significantly reduced IL-1β, IL-6, and TNF-α protein levels induced by an oral bacterial pathogen *Aa* compared with controls. In this study, we hypothesize that pharmacological inhibition of S1PR2 by JTE013 will inhibit IL-1β, IL-6, and TNF-α protein levels induced by *Aa* compared with vehicle controls. BMMs derived from C57BL/6J mice were treated for 30 min with either vehicle (ethanol) or JTE013 (2 to 8 µM). Then, BMMs were either uninfected or infected with *Aa* (1.5 CFU/cell) for 6 h. As shown in Figure 1A–C, vehicle or JTE013 treatment did not initiate significant proinflammatory cytokine production. Bacterial infection significantly induced the generation of IL-1β, IL-6, and TNF-α in BMMs treated with vehicle. In contrast, treatment with JTE013 dose-dependently decreased IL-1β, IL-6, and TNF-α levels induced by *Aa*. JTE013 (8 µM) reduced IL-1β by 90.2%, IL-6 by 80.8%, and TNF-α by 49.5% induced by *Aa* compared with the vehicle-treated controls. Bacterial infection significantly increased S1P by 21.5% in cell protein lysates (Figure 1D). JTE013 (8 µM) significantly reduced S1P levels by 45.5% in cell protein lysates of BMMs without *Aa* infection and by 57.9% in cell protein lysates of BMMs with *Aa* infection (Figure 1D). The levels of S1P in cell culture media were below detection level (data not shown). There was no significant cell toxicity observed on BMMs 24 h after treatment with JTE013 (2 to 8 µM, Figure 1E).

### 3.2. Inhibition of S1PR2 by JTE013 Reduced p-PI3K, p-ERK, p-JNK, p-p38, and p-NF-kBp65 Protein Expressions Induced by Aa in BMMs

Our previous study [23] showed that knockdown of S1PR2 by the S1PR2 shRNA lentiviral vector suppressed p-PI3K, p-ERK, p-JNK, p-p38, and p-NF-κBp65 protein expressions 4 h after infection with *Aa*. To further elucidate which signaling pathways affected by JTE013 in regulating the immune response induced by *Aa,* we performed western blot analysis of BMMs treated either with vehicle or JTE013 (8 µM) with or without *Aa* infection for 4 h. As shown in Figure 2A–F, bacterial infection significantly increased the p-PI3K, p-ERK, p-JNK, p-p38, and p-NF-κBp65 protein levels. JTE013 inhibited p-PI3K by 85.5%, p-ERK by 96.1%, p-JNK by 58.4%, p-p38 by 35.7%, and p-NF-κBp65 by 81.8% in BMMs infected with *Aa* compared with vehicle-treated cells infected with *Aa*. These results support that S1PR2 controls PI3K, ERK, JNK, and p38, and NF-κB signaling pathways, which contribute to proinflammatory cytokine release induced by the oral pathogen *Aa*.

### 3.3. S1PR2 Regulates Cell Chemotaxis Induced by Aa-Stimulated Cell Culture Media

Proinflammatory cytokines and S1P initiate the chemotaxis response of cells. Since inhibition of S1PR2 by JTE013 significantly reduced IL-1β, IL-6, TNF-α, and S1P levels induced by *Aa,* we hypothesize that S1PR2 inhibition will attenuate BMMs chemotaxis induced by *Aa*-stimulated cell culture media. Figure 3 shows that BMMs treated with the S1PR2 shRNA and incubated with *Aa*-stimulated media (derived from BMMs treated with S1PR2 shRNA and infected with *Aa* for 6 h) reduced cell migration by 2.3-fold compared with BMMs treated with control shRNA and incubated with *Aa*-stimulated media (derived from BMMs treated with control shRNA and infected with *Aa* for 6 h). BMMs treated with the S1PR2 shRNA decreased the S1PR2 mRNA levels by 65.2% compared with control shRNA treatment (data not shown). Additionally, BMMs treated with JTE013 (8 µM) and incubated with *Aa*-stimulated media (derived from BMMs treated with JTE013 and infected with *Aa* for 6 h) reduced cell chemotaxis by 3.3-fold compared with BMMs treated with vehicle and incubated with *Aa*-stimulated media (derived from BMMs treated with vehicle and infected with *Aa* for 6 h). 

### 3.4. Inhibition of S1PR2 by JTE013 Suppressed Osteoclastogenesis and Bone Resorption In Vitro

To test the efficacy of the S1PR2 antagonist (JTE013) in osteoclastogenesis, we performed an osteoclastogenesis assay in BM cells treated with either vehicle or JTE013 (8 µM). As shown in Figure 4A, there were no TRAP^+^ osteoclasts in cells treated without stimulation or treated only with *Aa*-stimulated cell culture media. There were many TRAP^+^ multinucleated osteoclasts in cells treated with vehicle stimulated with RANKL alone or co-cultured with both RANKL and *Aa*-stimulated cell culture media. In contrast, JTE013 treatment significantly decreased both size and number of TRAP^+^ multinucleated osteoclasts compared with vehicle control groups (Figure 4A–C). JTE013 reduced the number of osteoclasts by 95.3% in BM cells stimulated with RANKL, and by 83.7% in cells co-cultured with both RANKL and *Aa*-stimulated cell culture media compared with the control vehicle groups. JTE013 also reduced the area of osteoclasts by 99.1% in BM cells stimulated by RANKL alone, and by 88.7% in BM cells co-cultured with both RANKL and *Aa*-stimulated cell culture media compared with the control vehicle groups. 

We further determined the efficacy of JTE013 in bone resorbing activity induced by RANKL. Figure 4D shows that there were no bone resorption pits without stimulation or cells cultured with only *Aa*-stimulated cell culture media. RANKL alone or co-cultured with RANKL and *Aa*-stimulated cell culture media induced the formation of bone resorption pits in BM cells treated with vehicle. In contrast, JTE013 (8 µM) completely inhibited bone resorption in BM cell cultures in the presence of RANKL with or without co-culture with *Aa*-stimulated cell culture media.

### 3.5. Inhibition of S1PR2 by JTE013 Attenuated the mRNA Levels of Nfatc1, Ctsk, Acp5, Oscar, Dc-Stamp, and Oc-Stamp Induced by RANKL

Our previous study [23] demonstrated that knockdown of S1PR2 by a specific S1PR2 shRNA significantly reduced the mRNA levels of osteoclastogenic genes—including Nfatc1, Ctsk, Acp5, Oscar, Dc-stamp, and Oc-stamp—in total BM cells three days after RANKL treatment. To further determine the role of the S1PR2 antagonist (JTE013) on the expressions of RANKL-induced osteoclast associated genes, we performed an osteoclastogenesis assay and used real-time PCR to determine these osteoclastogenic gene expressions three days after RANKL treatment. As shown in Figure 5, *Aa*-stimulated cell culture media did not induce any significant changes in these osteoclastogenic genes. Treatment with RANKL or co-culture of cells with both RANK and *Aa*-stimulated cell culture media significantly enhanced the mRNA levels of these osteoclastogenic genes. In contrast, JTE013 (8 µM) significantly inhibited the mRNA levels of these osteoclastogenic genes induced by RANKL with or without co-culture with *Aa*-stimulated cell culture media (Figure 5A–F). JTE013 reduced Nfatc1 by 62.3%, Ctsk by 60.0%, Acp5 by 71.4%, Oscar by 74.2%, Dc-stamp by 86.0%, and Oc-stamp by 76.4% in BM cells treated with RANKL compared with vehicle controls. JTE013 also decreased Nfatc1 by 57.8%, Ctsk by 43.7%, Acp5 by 66.4%, Oscar by 62.0%, Dc-stamp by 74.5%, and Oc-stamp by 74.9% in BM cells co-cultured with both RANKL and *Aa*-stimulated cell culture media compared with vehicle controls. There were no significant differences of RANKL mRNA levels between vehicle and JTE013 groups among all the treatment groups (Figure 5G). In contrast, JTE013 reduced OPG by 88.3% in BM cells co-stimulated by RANKL and *Aa*-stimulated media compared with vehicle control group (Figure 5H). 

### 3.6. S1PR2 Controls the Activation of Podosome Protein Kinases PI3K, Src, and Pyk2 Induced by RANKL with or without Co-Stimulation by Aa-Stimulated Media

Osteoclastogenesis involves adhesion, followed by fusion of monocytes and macrophages [26]. Thus, we further analyzed the role of S1PR2 on podosome components, PI3K, Src, and Pyk2, which are protein kinases contributing to adhesion and fusion of BMMs [27]. Three days after lentiviral infection, the S1PR2 shRNA reduced the S1PR2 mRNA level by an average of 63.9% compared with control shRNA treatment (Figure 6A). The BMMs were unstimulated, cultured with RANKL, or co-cultured with RANKL and *Aa*-stimulated cell culture media for 1 h. As shown in Figure 6B–I, there were no significant differences in p-PI3K, p-Src, or p-Pyk2 protein levels in BMMs without stimulation between control shRNA versus S1PR2 shRNA treatment and vehicle versus JTE013 treatment. Treatment with RANKL or co-treatment with both RANKL and *Aa*-stimulated media significantly increased p-PI3K, p-Src, and p-Pyk2 in BMMs treated with either control shRNA or vehicle. In contrast, knockdown of S1PR2 by S1PR2 shRNA or inhibition of S1PR2 by JTE013 significantly reduced these protein kinase levels induced by RANKL alone or by co-culture by RANKL and *Aa*-stimulated cell culture media. The S1PR2 shRNA treatment decreased the average of protein levels of p-PI3K by 53.9%, p-Src by 40.6%, and p-Pyk2 by 58.2%, respectively, in BMMs stimulated by RANKL compared with control shRNA treatment. The S1PR2 shRNA treatment attenuated the average protein levels of p-PI3K. by 51.7%, p-Src by 34.9%, and p-Pyk2 by 43.6%, respectively, in BMMs co-cultured with RANKL and *Aa*-stimulated cell culture media. JTE013 (8 µM) decreased the average protein levels of p-PI3K by 52.9%, p-Src by 45.5%, and p-Pyk2 by 49.1%, respectively, in BMMs stimulated by RANKL compared with control vehicle treatment. JTE013 (8 µM) also attenuated the average protein levels of p-PI3K by 54.3%, p-Src by 30.7%, and p-Pyk2 by 35.1%, respectively, in BMMs co-cultured with RANKL and *Aa*-stimulated cell culture media. 

### 3.7. S1PR2 Regulates Podosome Adhesive Protein F-Actin, Integrin β_3,_ and Paxillin Levels Induced by RANKL with or without Co-Stimulation by Aa-Stimulated Media

Podosome adhesive proteins (including F-actin, integrin β_3,_ and paxillin) play a critical role in cell adhesion and fusion during osteoclastogenesis. Since knockdown of S1PR2 by S1PR2 shRNA [23] or inhibition of S1PR2 by JTE013 (Figure 4) suppressed osteoclastogenesis and bone resorption induced by RANKL, we hypothesize that S1PR2 regulates podosome adhesive protein levels induced by RANKL. After treatment with RANKL for three days, monocytes and macrophages will differentiate into pre-osteoclasts and subsequently will fuse together to form multinucleated osteoclasts. Our previous studies showed that the levels of F-actin, integrin β_3_, and paxillin did not increase significantly until three days after RANKL treatment (when the cells become pre-osteoclasts, data not shown). We use both the S1PR2 shRNA and S1PR2 specific antagonist (JTE013, 8 µM) to determine the role of S1PR2 in podosome adhesive protein expressions induced by RANKL. As shown in Figure 7A–H, there were no significant differences in F-actin, integrin β_3_, and paxillin protein levels in BMMs without stimulation between control shRNA versus S1PR2 shRNA or vehicle versus JTE013 treatment. Treatment with RANKL or co-treatment with both RANKL and *Aa*-stimulated media significantly increased F-actin, integrin β_3_, and paxillin protein levels in BMMs treated with either control shRNA or vehicle. In contrast, knockdown of S1PR2 by S1PR2 shRNA or inhibition of S1PR2 by JTE013 significantly reduced these adhesive protein levels induced by RANKL alone or by co-culture of RANKL and *Aa*-stimulated cell culture media. S1PR2 shRNA decreased the average of protein levels of F-actin by 49.1%, integrin β_3_ by 62.3%, and paxillin by 59.8%, respectively, in BMMs stimulated by RANKL compared with control shRNA treatment. Additionally, S1PR2 shRNA decreased the average of protein levels of F-actin by 49.7%, integrin β_3_ by 65.6%%, and paxillin by 56.3% induced by both RANKL and *Aa*-stimulated media compared with control shRNA treatment. Similarly, JTE013 reduced the average of protein levels of F-actin by 57.8%, integrin β_3_ by 69.4%, and paxillin by 54.0% induced by RANKL compared with vehicle control groups. Additionally, JTE013 decreased the average of protein levels of F-actin by 66.1%, integrin β_3_ by 50.8%, and paxillin by 69.2% induced by both RANKL and *Aa*-stimulated media compared with vehicle controls. 

Next, we used immunofluorescence to determine the role of S1PR2 on podosome adhesive protein expression. As shown in Figure 8, there were only low levels of F-actin, integrin β_3,_ and paxillin in BMMs treated with either control shRNA, S1PR2 shRNA, vehicle, or JTE013 without stimulation. In BMMs treated with control shRNA or vehicle and cultured with RANKL alone, or co-cultured with both RANKL and *Aa*-stimulated media, there were increased levels of F-actin (mainly localized at the plasma membrane of mononucleated cells and some in the cytoplasm of multinucleated cells), integrin β_3_ (mainly in the cytoplasm of mononucleated cells), and paxillin (mainly in the cytoplasm of both mononucleated cells and multinucleated cells). F-actin was co-localized with integrin β_3_ or paxillin in these cells. In contrast, treatment with S1PR2 shRNA or JTE013 suppressed F-actin, integrin β_3_, and paxillin expressions induced by RANKL alone or by both RANKL and *Aa*-stimulated media. Fluorescence intensity analysis (Figure 8E–J) revealed that treatment with RANKL or co-treatment with both RANKL and *Aa*-stimulated media significantly increased the fluorescence intensity of F-actin, integrin β_3_, and paxillin. The S1PR2 shRNA treatment decreased the average fluorescence intensity of F-actin by 35.9%, integrin β_3_ by 42.5%, and paxillin by 36.6%, respectively, in BMMs stimulated by RANKL compared with control shRNA treatment. Additionally, S1PR2 shRNA decreased the average fluorescence intensity of F-actin by 28.6%, integrin β_3_ by 52.5%, and paxillin by 36.6%, respectively, induced by both RANKL and *Aa*-stimulated media compared with control shRNA treatment. Similarly, JTE013 reduced the average fluorescence intensity of F-actin by 32.7%, integrin β_3_ by 51.9%, and paxillin by 39.0% induced by RANKL compared with vehicle control groups. Additionally, JTE013 decreased the average fluorescence intensity of F-actin by 30.9%, integrin β_3_ by 57.5%, and paxillin by 39.9% induced by both RANKL and *Aa*-stimulated media compared with vehicle controls. 

## 4. Discussion

In this study, we demonstrated that pharmacological inhibition of S1PR2 by JTE013 inhibited IL-1β, IL-6, TNF-α, and S1P production induced by an oral bacterial pathogen; suppressed BMMs chemotaxis; and attenuated osteoclastogenesis and bone resorption induced by RANKL. Mechanistically, we demonstrated that S1PR2 regulated the PI3K, MAPKs, and NF-κB signaling cascades induced by the oral bacterial pathogen, as well as modulated the expressions of the podosome protein kinases (PI3K, Src, and Pyk2) and the adhesive proteins (F-actin, integrins, and paxillin) induced by RANKL, subsequently affecting osteoclastogenesis and bone resorption. Our study is the first to demonstrate that S1PR2 not only regulated osteoclastogenesis and bone resorption induced by S1P, but also could be initiated by bacterial pathogens and RANKL.

Mammalian cell membrane contains specialized membrane domains, called lipid rafts, which are enriched in cholesterol, glycosphingolipids, and proteins. Lipids rafts serve as signaling platforms that recruit transmembrane and intracellular signaling molecules, facilitating the interaction of these signaling molecules and supporting signaling transduction [32,33]. Proteins associated with the lipid rafts include glycosylphosphatidylinositol (GPI)-anchored proteins, heterotrimeric G proteins, Src family proteins, PI3K, integrins, and MAPKs [32,33]. It has been shown that oral pathogen *Aa* mainly initiates cytokine responses via toll like receptor (TLR)4 [34] and MyD88 is an adaptor protein for TLR4 on the host cell membrane [35]. RANKL binds with RANK on plasma membrane and TRAF6 is an adaptor for RANK [36]. Previous studies showed that TLR4 and MyD88 were recruited to the lipid rafts in response to oxidase stress [37] and fatty acid treatment [38]. Additionally, RANK and TRAF6 were co-localized in lipid rafts after RANKL stimulation [39]. Our studies suggested that S1PR2 might interact with TLRs, PI3K, integrins, Src, MAPKs, RANK, and TRAF6 molecules in lipid rafts, modulating immune signaling cascades. Future studies will need to determine if S1PR2 can co-localize with TLR4, MyD88, RANK, TRAF6, MAPKs, PI3K, Src, and integrins in lipid rafts in response to bacterial infection or RANKL stimulation. 

Previously, there were conflicting results regarding how S1PR2 regulates chemotaxis of macrophages in response to S1P [25,40]. Yang et al. [40] demonstrated that knockdown of S1PR2 by a S1PR2 siRNA or inhibition of S1PR2 by JTE013 suppressed S1P-induced cell chemotaxis. In contrast, Ishii et al. [25] showed that treatment with a S1PR2 siRNA increased S1P-induced cell chemotaxis of BMMs. In a bile duct ligation-induced liver injury animal model, Yang et al. [40] demonstrated that treatment with JTE013 in mice significantly reduced IL-6, TNF-α, interferon-γ (IFN-γ), monocyte chemoattractant protein-1(MCP-1) levels, and inhibited recruitment of inflammatory cells in the liver. However, Yang et al. [40] did not show S1P levels in the liver tissues of JTE013-treated versus vehicle-treated animals. Our study is consistent with the Yang et al. [40] finding that JTE013 reduced proinflammatory cytokines, subsequently affecting cell chemotaxis. The reduction of cell chemotaxis in our study was mainly caused by the reduced inflammatory cytokine levels in the cell culture media in cells treated with either S1PR2 shRNA or JTE013 compared with their controls. Although we did not detect S1P levels in the cell culture media, the reduction of S1P by JTE013 in cell protein lysates suggested that the S1P levels in the cell culture media might also be reduced by the JTE013 treatment.

Previously, Ishii et al. [25] showed that mice treated with JTE013 alleviated osteoporosis induced by either intraperitoneal injection of RANKL or by ovariectomy. They claimed that the effect of JTE013 was caused by inhibition of cell migration of monocytes from blood circulation (with high level of S1P) to bone tissues (with low level of S1P), subsequently affecting osteoclastogenesis [25]. Another study performed by Kikuta et al. [41] demonstrated that treatment with an active form of vitamin D or its analog, eldecalcitol, in vivo suppressed S1PR2 expression in circulating monocytes and alleviated ovariectomy-induced osteoporosis [41]. The Kikuta et al. [41] study supports the important role of S1PR2 in regulating bone loss. However, the authors explained that this effect was caused by vitamin D suppressing the migration of monocytes from the bone to the blood circulation [41]. We demonstrated for the first time that S1PR2 regulated podosome components (PI3K, Src, Pyk2, F-actin, integrins, and paxillin) induced by RANKL, subsequently affecting adhesion and fusion of osteoclasts. In the early stages of osteoclastogenesis, as shown in the study, we observed co-localization of F-actin with integrins and F-actin with paxillin on the plasma membrane of mononucleated cells and some co-localization in the cytoplasm of multinucleated cells. In the late stages of osteoclastogenesis, these podosome adhesive proteins were co-localized in a sealing zone structure [29]. The sealing zone structure creates an isolated resorption compartment between osteoclasts and bone matrix, which transports protons and proteolytic enzymes into the resorption compartment to dissolve minerals and degrade bone matrix protein [28,29,30]. As a result, reduction of podosome components by knockdown or inhibition of S1PR2 not only inhibited adhesion and fusion of osteoclasts induced by RANKL, but also suppressed bone resorption. In this study, we only demonstrated the effects of the S1PR2 inhibitor (JTE013) in modulating IL-1β, IL-6, TNF-α, and S1P production, cell chemotaxis, osteoclastogenesis, and bone resorption in vitro. Future studies will need to determine whether JTE013 can alleviate bone loss either induced by oral pathogens in a periodontitis animal model or induced by an autoimmune response in a rheumatoid arthritis animal model.

It was noted that there were conflicting results regarding how S1PR2 regulated bone homeostasis using different strains of S1PR2 deficient mice [25,42]. Ishii et al. [25] showed that S1PR2 deficient mice (*S1pr2^tm1Rlp^*) developed by Richard L Proia increased bone volume, numbers of trabecular bone, and trabecular thickness compared with wild type mice. In contrast, Weske et al. [42] reported that S1PR2 deficient mice (*S1pr2 ^tm1Jch^*) developed by Jerold Chun reduced bone volume, numbers of trabecular bone, and trabecular thickness compared with wild type mice. We obtained both strains of these mutant mice. Our in vitro studies revealed that the levels of S1PR2 mRNA can be detected in the BMMs derived from either mutant *S1pr2^tm1Rlp^* mice or mutant *S1pr2 ^tm1Jch^* mice, although the mutant mice had significantly lower S1PR2 mRNA levels compared with wild type mice (data not shown). Moreover, BMMs derived from either mutant *S1pr2^tm1Rlp^* mice or mutant *S1pr2 ^tm1Jch^* mice infected with the oral pathogen *Aa* generated significantly higher IL-1β, IL-6, and TNF-α compared with their wild type controls (data not shown), which is contradictory to our in vitro studies using either S1PR2 shRNA approach [23] or S1PR2 inhibitor (JTE013, Figure 1A–C). Furthermore, we observed different RANKL-induced osteoclastogenic responses using BM cells derived from these mutant mice. BM cells derived from the mutant *S1pr2^tm1Rlp^* mice displayed a reduction in RANKL-induced osteoclastogenesis compared with wild type controls (data not shown). In contrast, there were no significant differences in RANKL-induced osteoclastogenesis between the BM cells derived from the mutant *S1pr2 ^tm1Jch^* mice and wild type controls (data not shown). Future studies will need to determine whether these mutant S1PR2 mice exhibit truncated S1PR2 proteins, resulting non-specific target effects.

A recent study performed by Weske et al. [42] showed that conditional deleting or pharmacologically inhibiting S1P lyase increased bone formation in mice. It is known that S1P has multifaceted roles in bone remodeling, including regulation of the proliferation and survival of osteoblasts [43]. The role of S1P in bone homeostasis might be dependent on the S1P concentrations. In pathological conditions (periodontitis, osteoporosis, and rheumatoid arthritis), a significantly high S1P level induced by immune responses might favor a bone loss response. In physiological conditions, a mild increase of S1P might result in proliferation and survival of osteoblasts. Weske et al. [42] demonstrated that S1P increased OPG, nuclear accumulation of β-catenin, and non-canonical WNT5A-LRP5 signaling, promoting osteoblast differentiation [42]. Treatment with JTE013 in MC3T3-E1 cells blocked OPG production, nuclear β-catenin, and LRP5 expressions induced by S1P [42]. In our laboratory, treatment with JTE013 increased alkaline phosphatase and alizarin red staining (osteogenic markers) in BM-derived mesenchymal stromal cells cultured in osteogenic media compared with vehicle controls (data not shown). Future studies will need to determine how S1PR2 regulates osteoblast differentiation and bone formation.

## 5. Conclusions

In this study, we demonstrated that inhibition of S1PR2 by its specific antagonist (JTE013) reduced IL-1β, IL-6, TNF-α, and S1P production induced by the oral bacterial pathogen *Aa*; inhibited BMMs chemotaxis induced by bacterial infection; as well as suppressed osteoclastogenesis and bone resorption induced by RANKL. ShRNA knockdown of S1PR2 or inhibition of S1PR2 by JTE013 suppressed podosome components, including PI3K, Src, Pyk2, integrin β_3,_ F-actin, and paxillin levels induced by RANKL in murine bone marrow cells. Our results suggested that inhibition of S1PR2 by its specific antagonist (JTE013) could be a good strategy to treat bone loss associated with skeletal diseases.

## Figures and Tables

**Figure 1 cells-08-00017-f001:**
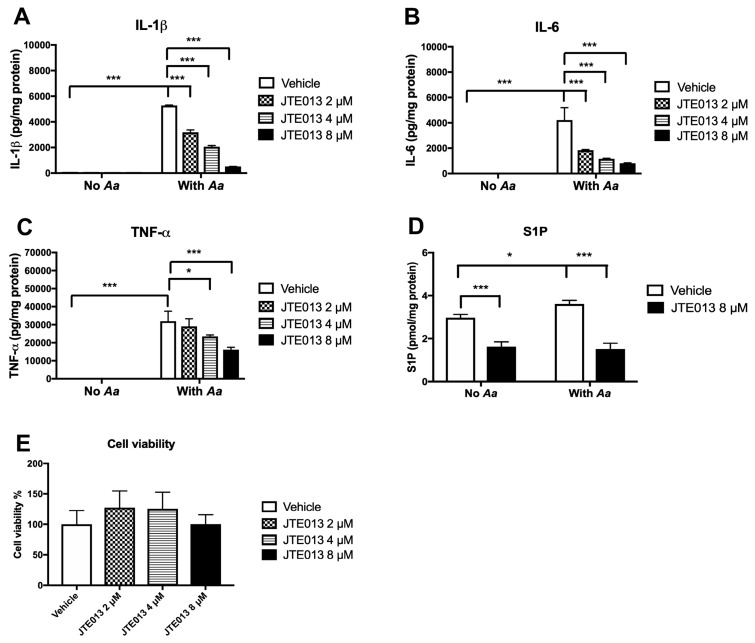
Inhibition of S1PR2 by its specific antagonist (JTE013) significantly decreased IL-1β, IL-6, TNF-a, and S1P levels induced by *Aa* in BMMs. Murine BMMs were treated either with vehicle or JTE013 for 30 min. Then cells were either uninfected or infected with *Aa* for 6 h. (**A**) IL-1β, (**B**) IL-6, and (**C**) TNF-α levels were quantified by ELISA. (**D**) S1P levels in cell protein lysates were quantified by mass spectrometry. Cytokine and S1P levels were normalized by protein levels in cell lysates. (**E**) Cell viability in BMMs treated with vehicle or JTE013 for 24 h. (* *p* < 0.05 *** *p* < 0.001).

**Figure 2 cells-08-00017-f002:**
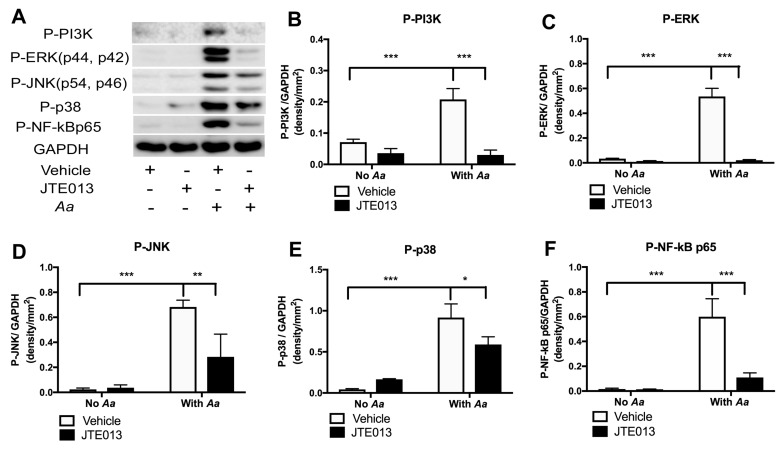
Inhibition of S1PR2 by its specific antagonist (JTE013) attenuated p-PI3K, p-ERK, p-JNK, p-p38, and p-NF-kBp65 protein expressions induced by *Aa* in BMMs. Murine BMMs were treated with either vehicle or JTE013 for 30 min. Then cells were either uninfected or infected with *Aa* for 4 h. (**A**) p-PI3K, p-ERK, p-JNK, p-p38, p-NF-κBp65, and control GPADH protein expressions were evaluated by western blot. (**B**) p-PI3K protein density, (**C**) p-ERK protein density, (**D**) p-JNK protein density, (**E**) p-p38 protein density, and (**F**) p-NF-κBp65 protein density were analyzed and normalized by GAPDH protein expression. (* *p* < 0.05, ** *p* < 0.01, *** *p* < 0.001).

**Figure 3 cells-08-00017-f003:**
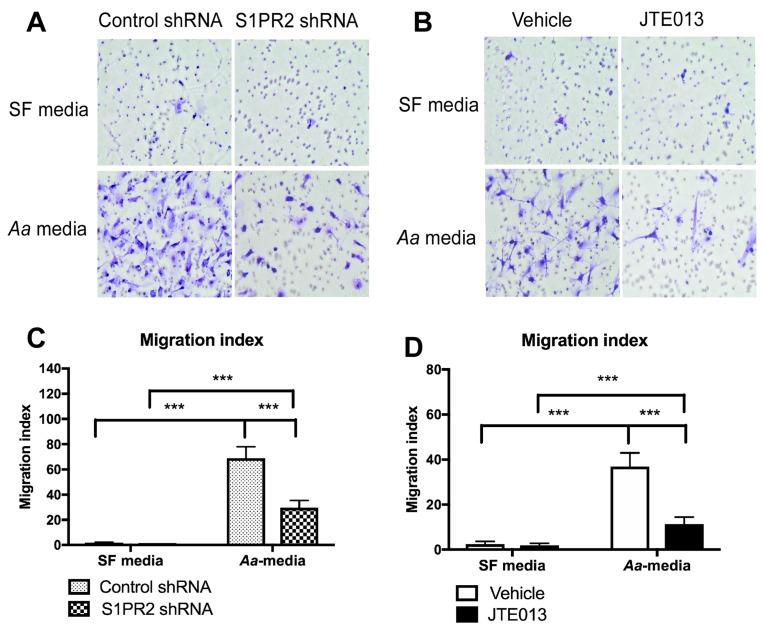
S1PR2 regulates cell chemotaxis induced by *Aa*-stimulated cell culture media. Murine BMMs, treated with either S1PR2 shRNA, control shRNA, JTE013, or vehicle, respectively, were loaded on the top inserts of transwell plates. The bottom chambers were filled with either serum-free (SF) media or *Aa*-stimulated cell culture media (*Aa-media*) as described in Methods. Cells were incubated for 6 h. (**A**,**B**) Representative images show crystal violet staining of cells on the bottom of inserts. (**C**,**D**) display migration index. (*** *p* < 0.001).

**Figure 4 cells-08-00017-f004:**
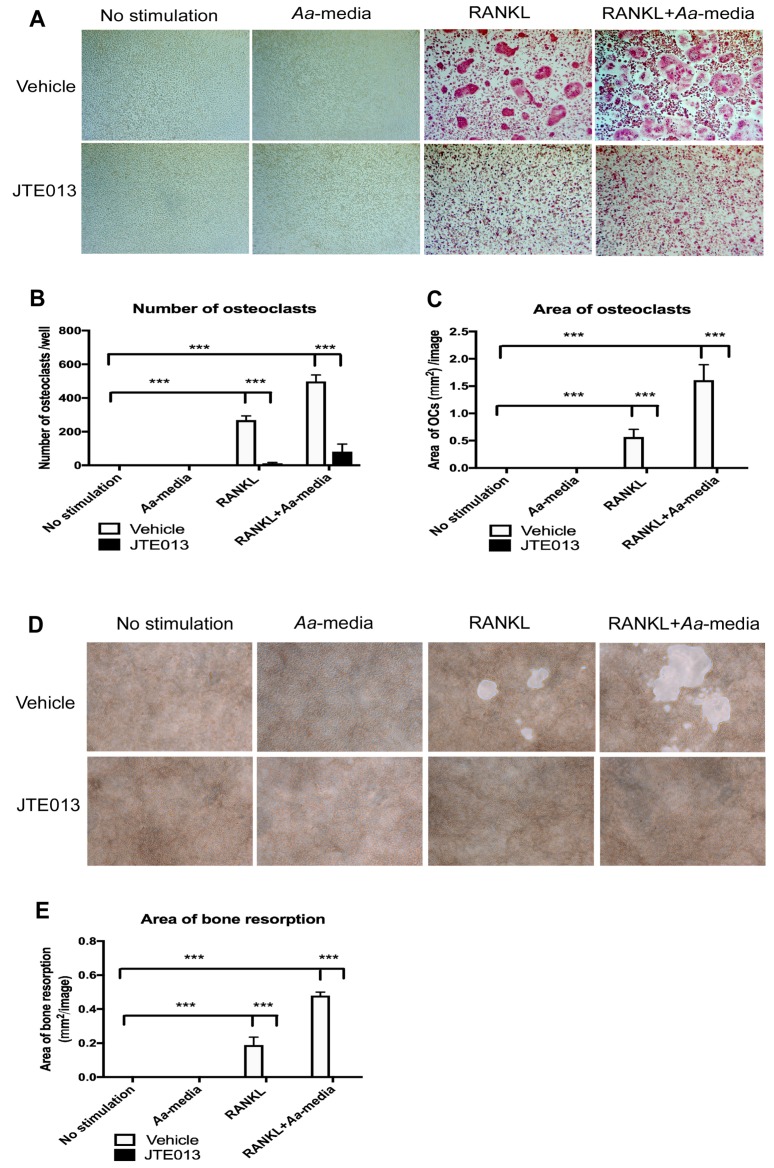
Inhibition of S1PR2 by its specific antagonist (JTE013) suppressed osteoclastogenesis and bone resorption induced by RANKL with or without co-culture with *Aa*-stimulated cell culture media. Murine BMs were treated with vehicle or JTE013. Cell were either unstimulated, cultured with *Aa*-stimulated cell culture media (*Aa*-media) alone, cultured with RANKL alone, or co-cultured with both RANKL and *Aa*-media as described in Methods. (**A**) Representative images show TRAP-stained cells at the 100× magnification view. (**B**) Number of TRAP^+^ multinucleated (more than three nuclei) osteoclasts/well (96-well) and (**C**) Total areas of osteoclasts/image were quantified. (**D**) Representative images show bone resorption pits at 100× magnification view. (**E**) Total areas of bone resorption pits /image were quantified. (*** *p* < 0.001).

**Figure 5 cells-08-00017-f005:**
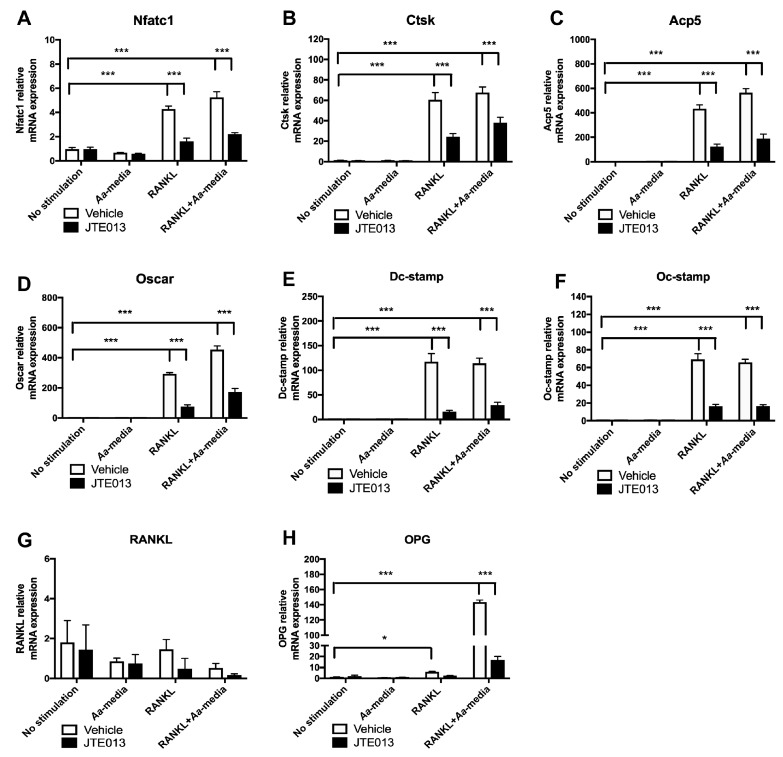
Inhibition of S1PR2 by its specific antagonist (JTE013) attenuated Nfatc1, Ctsk, Acp5, Oscar, Dc-stamp, and Oc-stamp mRNA expressions induced by RANKL with or without co-culture with *Aa*-stimulated cell culture media. Murine BMs were treated with vehicle or JTE013. Cell were either unstimulated, cultured with *Aa*-stimulated cell culture media (*Aa*-media) alone, RANKL alone, or co-cultured with both RANKL and *Aa*-media as described in Methods. (**A**) Nfatc1 mRNA, (**B**) Ctsk mRNA, (**C**) Acp5 mRNA, (**D**) Oscar mRNA, (**E**) Dc-stamp mRNA, (**F**) Oc-stamp mRNA, (**G**) RANKL mRNA, and (**H**) OPG mRNA levels were quantified by real time PCR and normalized by GAPDH expression. (* *p* < 0.05, *** *p* < 0.001).

**Figure 6 cells-08-00017-f006:**
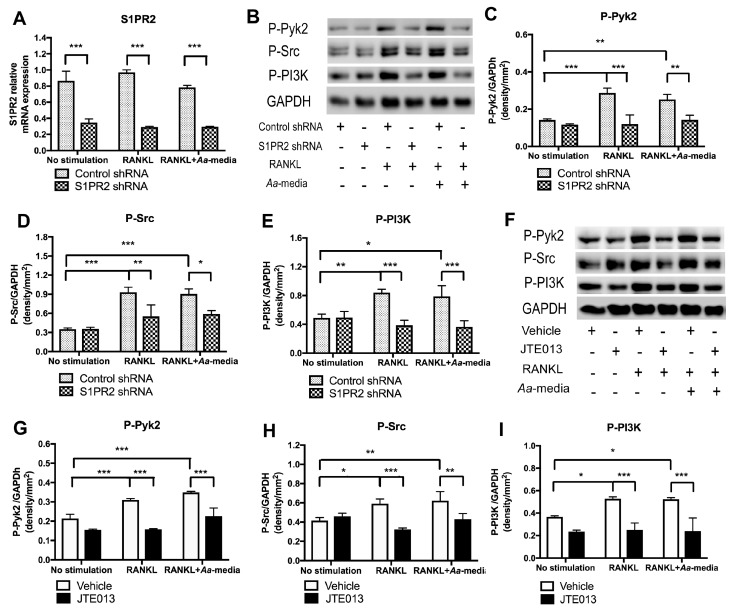
S1PR2 controls the activation of podosome protein kinases PI3K, Src, and Pyk2 induced by RANKL with or without co-stimulated by *Aa*-stimulated media. Murine BMMs were treated with control shRNA/S1PR2 shRNA for three days or treated with vehicle/ JTE013 for 30 min. Cells were unstimulated, stimulated with RANKL, or co-stimulated with both RANKL and *Aa*-stimulated media (*Aa*-media) for 1h. (**A**) S1PR2 mRNA levels were quantified by real time PCR and normalized by GAPDH expression. (**B**) and (**F**) p-Pyk2, p-Src, p-PI3K, and control GPADH protein expressions were evaluated by western blot. (**C**) and (**G**) p-Pyk2 protein density, (**D**) and (**H**) p-Src protein density, and (**E**) and (**I**) p-PI3K protein density were analyzed and normalized by GAPDH protein expression. (* *p* < 0.05, ** *p* < 0.01, *** *p* < 0.001).

**Figure 7 cells-08-00017-f007:**
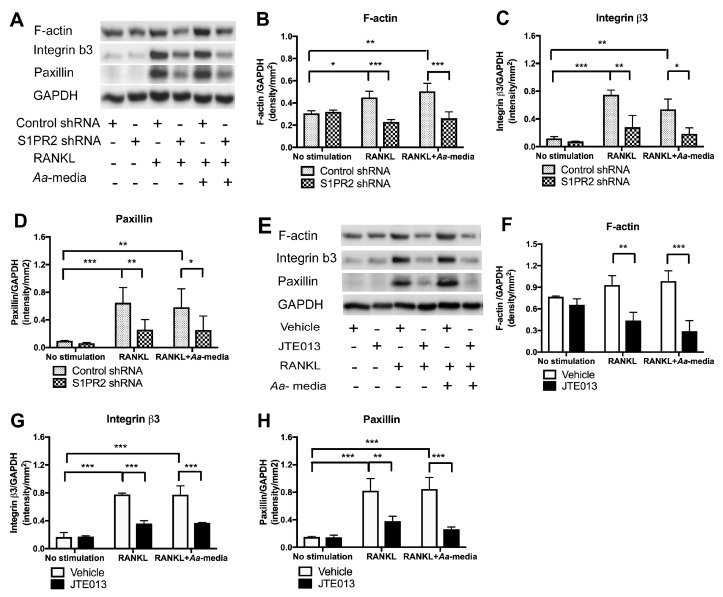
S1PR2 regulates podosome adhesive protein F-actin, integrin β_3_, and paxillin levels induced by RANKL with or without co-stimulated by *Aa*-stimulated media. Murine BMMs were treated with control shRNA/S1PR2 shRNA for one day or treated with vehicle/ JTE013 for 30 min. Then, cells were cultured in media containing M-CSF (50 ng/mL) alone or M-CSF with RANKL (250 ng/mL) for two days. On the third day, the media was changed with or without RANKL and/or JTE013. Some of the cells were co-cultured with both RANKL and *Aa*-stimulated media (*Aa*-media, 200 µL/mL) for another day. (**A**,**E**) F-actin, integrin β3, paxillin, and GAPDH protein expressions were evaluated by western blot. (**B**,**F**) F-actin protein density, (**C**,**G**) integrin β3 protein density, and (**D**,**H**) paxillin protein density were analyzed and normalized by GAPDH protein expression. (* *p* < 0.05, ** *p* < 0.01, *** *p* < 0.001).

**Figure 8 cells-08-00017-f008:**
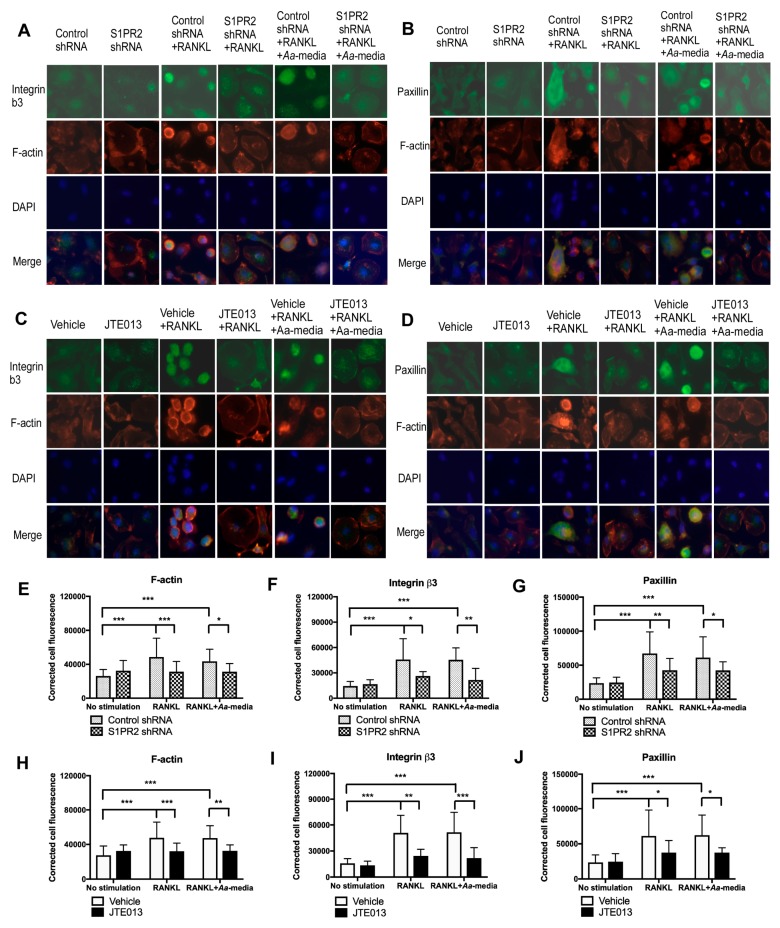
Immunofluorescence staining of integrin β3, F-actin, and paxillin in BMMs. Murine BMMs were treated with control shRNA/S1PR2 shRNA for one day or treated with vehicle/ JTE013 for 30 min. Then, cells were cultured in media containing M-CSF (50 ng/mL) alone or M-CSF with RANKL (250 ng/mL) for two days. On the third day, the media was changed with or without RANKL and/or JTE013. Some of the cells were co-cultured with both RANKL and *Aa*-stimulated media (*Aa*-media, 200 µL/mL) for another day. (**A**,**C**) show representative images of integrin β3, F-actin, and DAPI staining in BMMs (four to six cells per image). (**B**,**D**) show representative images of paxillin, F-actin, and DAPI staining in BMMs (four to six cells per image). (**E**,**H**) show cell fluorescence intensity of F-actin. (**F**,**I**) show cell fluorescence intensity of integrin β3. (**G**,**J**) show cell fluorescence intensity of paxillin. (* *p* < 0.05, ** *p* < 0.01, *** *p* < 0.001).

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
