# Peer review of "Sphingosine-1-Phosphate Receptor 2 Controls Podosome Components Induced by RANKL Affecting Osteoclastogenesis and Bone Resorption"

_cells, 2019, doi:10.3390/cells8010017_

Reviewer 1 Report

This manuscript describes that S1PR2 regulates IL-1β, IL-6, TNF-α, and S1P production, BMMs chemotaxis, osteoclastogenesis, and bone resorption. Author demonstrated that inhibition of S1PR2 by its specific antagonist JTE013 could be a good strategy to treat bone loss associated with inflammatory conditions.

This study is well written but has some major issues. The complete study is based on cell culture data and author clam that Sphingosine-1-phosphate receptor 2, a new target for inflammatory bone loss diseases. Author should demonstrate therapeutic role in in-vivo animal model or change title and most of the discussion part. Author previously, published paper in plos one in 2016 showing role of S1PR2 in osteoclast differentiation. Previously, Ishii et al (J Exp Med. 2010 Dec 20;207(13):2793-8.) used in-vivo model to show therapeutic role of JTE013 in bone loss. This manuscript does not add new to existing knowledge. Fig 6, 7 and 8 findings are novel. Author should re-write manuscript based on Fig 6, 7 and 8 (which is novel).

Minor issue: Introduction of manuscript is too complex.

Author Response

Please check the uploaded documents for response to reviewer's critiques. 

Reviewer 2 Report

The authors tested the effects of sphingosine-1-phosphate receptor 2 (S1PR2) inhibition on different pathways in monocytes and macrophages, and on their differentiation into osteoclasts responsible for bone resorption. They show that inhibition of S1PR2 significantly affects inflammation-related signaling and formation, maturation, and function of osteoclasts. The data are interesting, but the presentation must be improved. Also, some claims do not match the data shown.                      

1.     Lines 132-137. Indicate antibody dilutions used for Western.

2.     Fig. 5. The levels of mRNA do not necessarily reflect the levels of protein expression. To claim the reduction in protein levels the authors need to do Westerns, show the results and their quantification, as in Figs. 2, 6, and 7.

3.     Fig 8. The authors should specify how many cells were examined in each condition, or better yet, quantify the results using sufficient number of cells and appropriate statistics.

4.     Extensive editing is needed, preferably by a native speaker. E.g., line 33, “characterized with high levels” should be “characterized by high levels”; lines 44-45, Rac, Rho, phospholipase C (PLC), and NF-κB are not protein kinases; line 46, “MAPKs consist of” should be “ MAPKs include”; line 225, “expressions” should be “expression”; past tense should be used throughout; many sentences should be corrected to make them grammatical; etc.

Author Response

Please check the uploaded document for response for reviewer's critiques. 

Reviewer 3 Report

Hsu et al propose in this paper sphingosine 1-phosphate receptor 2 (S1PR2) as a new target for inflammatory bone diseases. The article is well written, straightforward with an intelligent design of experiments. Nevertheless, there is a lack of novelty in the message of the paper, as half of it is a replica from Hong Y, plos One, 2016 (the only difference is the use of antagonist JTE013 to block S1P2R instead of shS1PR2).

Major comments :

1. References from 16 to 19 are not correct, they are very old (2002) and not representative of S1P receptor metabolism. References from Kihara Y, Chun J or Spiegel S should be cited.

2. Authors used in this study S1PR2 antagonist JTE013 at 8 µM prepared in ethanol. Why did the authors chose this concentration ? did they check for effects on viability or proliferation of the antagonist in their cell models ? did they check for effects of the vehicule ethanol alone ?

Authors should demonstrate that JTE013 at 8 µM is the best concentration to block S1P signalization through S1P2 receptor, a dose-dependent effect has to be shown at least in one downstream marker.

3. It will be interesting to realize IHC experiments of podosomes with S1P2 antibody, check for a colocalization and eventually a direct interaction between S1PR2 and podosome components. Did authors realize IHC experiments with S1PR2 antibody ?

Minor comments :

1. Protocol 2.6 has different police sizes. Please correct.

2. Please change 5th in 5th in protocole 2.9

3. Graphs in figure 4 have not the same sizes, please modify.

4. Discussion need improvement. Some recent articles about the role of S1P metabolism in bone loss pathologies are lacking (Weske et al, Nat Med, 2018 for exemple). Please complete the discussion with the last relevant articles of the field.

Author Response

Please check the uploaded document for response to reviewer's critiques. 

Round  2

Reviewer 2 Report

The authors analyzed the effect of sphigosine-1phosphate receptor 2 on RANKL-induced osteoclastogenesis and bone resorption. They show that inhibition of S1PR2 significantly affects inflammation-related signaling and formation, maturation, and function of osteoclasts. The manuscript was significantly improved in revision, but several further improvements are necessary.

1.     The authors keep assuming that mRNA levels are directly proportional to protein expression. They should clarify language to reflect what is actually measured. While measurements of protein levels are not always feasible, the readers should not be misled by this assumption.

2.     Lines 406-7. Either show the data on the effect of JTE013 on sphingosine levels (possibly in the supplement), or, if the results are not statistically significant, do not mention them.

3.     Line 650. “The data were representatives from three separate experiments” should read “the data are presented as means +/- SD (or SEM, whatever the case may be) of three independent experiments”. Also, in this case n=3, not n=4, as stated.

4.     Similar corrections needed in other figure legends.

5.     Figs. 4,5,8. While the authors are perfectly justified in showing just a few typical cells, statistical analysis should be performed on sufficient number of cells to make it meaningful. This number should be indicated. Fig. 8 would be greatly improved by quantification and statistical analysis.

6.     Some editing is still needed. E.g., line 179 “antibodies used were diluted in 1:1000 condition” should read “antibodies were used at 1:1,000 dilution”; etc.

Author Response

Please see the attached response to reviewer's critiques. 

Reviewer 3 Report

Hsu et al propose has proposed an important revision of their paper. Nevertheless, I still have some comments and corrections to ask.

Line 65: Gi and Gq should be in index, pleas correct.

Line 72, JTE013 has to appear between brackets, please correct

Line 76, “the oral inhibitor” has to be corrected by “an oral inhibitor”

Line 406-407, the hypothesis that JTE013 could increase sphingosine content by inhibiting sphingosine kinase activity (1 or 2 or both) seems very risky and not validated. This fact will indicate that JTE013 antagonist is not specific. I suggest either measure SK1 or SK2 activities in the studied context to validate author’s hypothesis or erase the phrase.

Line 431, in the liver is written twice, one time is enough

Line 444, change “expresson” for expression

Line 469, change “developled” for developed

Line 489, change “cocentrations” for concentrations

Line 501, arrange the phrase “an oral bacterial….

Line 650, in the legend of Fig.1 there is an n=4 and after that it is written representatives from 3, please check this

Line 650, 658, 665, 675, 684, 693, 705, please change “the date are representatives from” for “the data are representative from”

Figure 4C/4E will be improved if µm2 is changed in mm2

All figures need to be checked, police are very different in the same figure and legends are repeated each time for each pannel. Sometimes, legends can be placed strategically between two or more pannels, figures could be lighten a little bit in this way.

In the conclusion part, « podosomes » word has to appear (is part of the title of the article)

Author Response

(The authors gave the same response as above.)
